# 3D-printed external cranial protection following decompressive craniectomy after brain injury: A pilot feasibility cohort study

**Karen Sui Geok Chua**[1]*, **Rathi Ratha Krishnan**[1], **Jia Min Yen**[1], **Tegan Kate Plunkett**[1], **Yan Ming Soh**[1‡], **Chien Joo Lim**[2], **Catherine M. Chia**[3], **Jun Cong Looi**[3‡], **Suan Gek Ng**[4‡], **Jai Rao**[4]

1 Tan Tock Seng Rehabilitation Centre, Tan Tock Seng Hospital, Singapore, Singapore, 2 Clinical Research and Innovation Office, Tan Tock Seng Hospital, Singapore, Singapore, 3 AuMed Pte Ltd, Singapore, Singapore, 4 Department of Neurosurgery, National Neuroscience Institute, TTSH Campus, Singapore, Singapore

☯ These authors contributed equally to this work.
‡ These authors also contributed equally to this work.
* Karen_Chua@ttsh.com.sg

**Data Availability Statement:** My full deidentified data points are available at the link below. https://www.ttsh.com.sg/Healthcare-Professionals/

## Abstract

### Objectives

3D-printed (3DP) customized temporary cranial protection solutions following decompressive craniectomy (DC) are currently not widely practiced. A pilot trial of a 3DP customized head protection prototype device (HPPD) on 10 subjects was conducted during the subacute rehabilitation phase.

### Materials and methods

Subjects > 30 days post-DC with stable cranial flaps and healed wounds were enrolled. HPPD were uniquely designed based on individuals' CT scan, where the base conformed to the surface of the individual's skin covering the cranial defect, and the lateral surface three-dimensionally mirrored, the contralateral healthy head. Each HPPD was fabricated using the fused deposition modeling method. These HPPD were then fitted on subjects using a progressive wearing schedule and monitored over 1, 2, 4, 6 and 8 follow-up (FU) weeks. Outcomes during FU included; reported wearing time/day (hours), subjective pain, discomfort, pruritus, dislodgment, cosmesis ratings; and observed wound changes. The primary outcome was safety and tolerability without pain or wound changes within 30 minutes of HPPD fitting.

### Results

In all, 10 enrolled subjects received 12 HPPDs [5/10 male, mean (SD) age 46 (14) years, mean (SD) duration post-DC 110 days (76)] and all subjects tolerated 30 minutes of initial HPPD fitting without wound changes. The mean (SD) HPPD mass was 61.2 g (SD 19.88). During 8 weeks of FU, no HPPD-related skin dehiscence was observed, while 20% (2/10) had transient skin imprints, and 80% (8/10) reported self-limiting pressure and pruritus.

Research-Innovation/Documents/Research/2019-00155/PONE_38286_DataPoints.xls.

**Funding:** KSGC, main author received competitive grant funding from the Tan Tock Seng Hospital Innovation fund in 2019 to fund this research work. (Grant number TIF_2019).

**Competing interests:** The authors have declared that no competing interests exist.

## Discussion

Findings from this exploratory study demonstrated preliminary feasibility and safety for a customized 3DP HPPD for temporary post-DC head protection over 8 weeks of follow-up. Monitoring and regular rest breaks during HPPD wear were important to prevent skin complications.

## Conclusion

This study suggests the potential for wider 3DP technology applications to provide cranial protection for this vulnerable population.

## Introduction

Decompressive craniectomy (DC), is an emergency and life-saving surgery performed following malignant cerebral infarctions, large intracerebral hemorrhages (ICH) or severe traumatic brain injury (TBI) [1]. A large section of skull is removed, underlying dura mater opened; damaged brain tissue is allowed to decompress and herniate through the skull defect, the scalp is then closed, without reimplantation of the bone. DC is associated with reduced intracranial pressures and improved mortality by halting the progression to permanent coma and brain death, however, a significant proportion survive with significant disability [2, 3].

DC may be associated with a variety of complications in ranging from 24.6% to 53.9% in early to subacute and chronic phases, being higher in those of older age and poor initial neurological status [4]. These include acute wound infections, paradoxical herniation of the cerebral cortex through the bone defect, subdural effusion, seizures, hydrocephalus and syndrome of the trephined brain [5–8].

Following DC, cranioplasty (CP), involving reconstruction of the excised cranium, with autologous bone, titanium or artificial materials, provides the only solution to these poorly understood complications. While CP improves rehabilitation from a motor and cognitive perspective, it may increase the possibility of postoperative complications, such as, hydrocephalous, seizures and infections [9].

The optimal timing of cranioplasty following initial DC remains controversial. Early CP (< 90 days of DC), compared to late CP (> 90 days DC) was more effective in improving motor functions compared to cognition or memory scores [10]. Early (< 12 weeks DC) compared to late (> 12 weeks DC) was not associated with significant group differences between specific complications, although late CP resulted in longer operating times related to bone and tissue flap dissection. However, significantly higher infection rates (P = 0.007) were found for CP performed < 14 days of initial DC, although the time period of 15–30 days post DC was associated with the lowest rate of bone resorption; the risk of new-onset seizures, which occurred only in patients who had undergone CP > 90 days after initial DC was increased [11].

In the first few weeks post DC, patients are identified to be at increased fall and injury risk due to impaired coordination and balance, thus fall prevention, cranial protection protocols and recommendations for mandatory helmet use during rehabilitative exercises are considered important. In spite of this, literature is scarce in regards to the optimal care and management of the DC skull defect [12–14]. A sole publication reported success in 4 patients, using an

invasive temporary method of methyl-methacrylate bone cement to immediately form an acrylic flap covering the DC craniectomy site [15].

While the importance for temporary head protection post-DC is agreed, no local standards of helmet use post-DC nor literature regarding acceptability or use. In the authors' anecdotal experience, local options include rigid, adjustable ventilated helmets, cycling helmets or semi-rigid sports helmets, all of which are poorly tolerated in the tropics due to helmet mass (250-300g each), skin occlusion and poor aesthetic appeal; leading to poor compliance and continued risk of an unprotected cranium till CP is performed.

The use of 3D technologies to customize cranial prostheses has been previously described. Cranial prostheses, whether implantable or externally applied, are generally designed to provide sufficient structural support and return of aesthetic features of the natural skull or head. Briefly, a cranial prosthesis design commonly begins with 3D information of the head, for instance CT scan, to provide the necessary 3D information of the defect. Often the contralateral skull or head is mirrored about the median plane of the patient to provide a design template for the contour of the prosthesis. When the defect outline and mirrored contour are used together, they form the basic 3D shape of a customized prosthesis.

Within the authors' limited knowledge, it was observed that few of the described methodologies used towards customized prostheses involved a parametric design approach, where a set a configuration is determined and used across multiple designs which each has unique dimensions and tolerances [16–18].

The authors further investigate the feasibility, safety and usability of such a parameterized approach in its application towards external customized protection, head protection prototype device (HPPD) for post-acute decompression craniectomy (DC) patients.

## Materials and methods

### (I) Study design

A pilot, single arm, feasibility study was conducted from 1 April 2019 to 30 September 2020. Institutional ethics board approval was granted by National Healthcare Group Domain Specific Review Boards. (NHG-DSRB 2019/00155). The study was registered with ClinicalTrials. gov (registration number: NCT-04021095) and the authors confirm that all ongoing and related trials for this intervention are registered. All subjects or their legally-appointed representatives gave written informed consent from prior to enrolment.

### (II) Study setting

The study and recruitment processes were conducted at the inpatient rehabilitation wards or outpatient rehabilitation clinics of the Tan Tock Seng Hospital (TTSH) Rehabilitation Centre, Singapore. The study sample were selected from consecutive inpatient admissions who had been previously screened by physiatrists at public hospitals during the acute stroke or TBI phase, and consecutive outpatients who were referred from acute hospital referral sources. The Functional Independence Measure (FIM), recorded within 72 hours of admission and discharge of rehabilitation by accredited rehabilitation therapists was used to characterize the patients' functional levels [19]. Prior to HPPD fabrication or fitting, routine rigid helmets were employed only in those with fully healed wounds following removal of surgical sutures, during supervised inpatient rehabilitation therapy sessions; examples included those provided here (https://www.danmarproducts.com) These helmets were removed once patients were enrolled.

## (III) Study participants

The inclusion criteria were: (i) age 21 to 80 years; (ii) unilateral decompressive craniectomy (DC), (iii) index conditions of acute ischaemic stroke (AIS), intracerebral haemorrhage (ICH), traumatic brain injury (TBI), subarachnoid haemorrhage (SAH), benign or malignant cerebral tumours, diagnosed by admitting neurologists or neurosurgeons, and confirmed on CT or MRI neuroimaging; (iv) > 30 days post-DC; (v) healed post-DC surgical wounds; (vi) presence of post-operative CT brain stereotaxis images; (vii) ability to understand simple instructions; and (viii) available carer/NOK for consent and assistance with donning of doffing of the HPPD and subjects' monitoring.

Subjects were excluded based on the presence of: (i) vegetative or minimally responsive states, in which disordered awareness, absent or inconsistent command-following or functional communication abilities were present; (ii) uncontrolled medical conditions (e.g. sepsis, delirium, active malignancy; (iii) end organ failure (end stage renal or liver failure, renal dialysis) or life expectancy < 6 months; (iv) pregnancy or lactation; (v) severe agitation or behavioural disturbance, active depression or anxiety, drug or alcohol addiction; (vi) unhealed or infected post-DC neurosurgical wounds, bulging or externally herniated cranial flaps; (vii) known allergy to the investigational products; (viii) unavailable post-DC CT brain imaging films needed for acquisition of the 3D printing images; (ix) lack of a carer who could assist with donning of doffing of the HPPD and monitoring for complications and compliance; unless the subject was independent in these.

## (IV) Study protocol

**(i) Design and fabrication of HPPD.** Firstly, a hemi-cranial customised design covering the deficient side of the DC, rather than a full protective helmet design was chosen to balance issues of mass and low tolerability with the need to provide cranial flap protection. Each of the subjects then underwent post-operative Cranium CT stereotaxis scans, at a slice thickness of 0.625mm. One subject at a time, anonymized data was imported in Digital Imaging and Communication in Medicine (DICOM, v3.0) format. The CT images were interpreted, and the cranium and cutaneous volume data were segmented and extracted using Mimics Innovation Suite (v22.0 Medical, Materialise NV, Belgium). Thereafter, the volume data was imported to 3-matic Medical (v14.0, Materialise NV, Belgium) as reference models for designing the HPPD. All important features of the HPPD were parameterized to ease the communication between the study and engineering teams, and to ensure uniform quality across all HPPDs. Briefly, the defect outline on the cranium was projected distally onto the cutaneous model. An empirically determined and optimized margin offset of 1.5 ± 0.5cm outward of the projected defect outline along the skin contour served as the base of the HPPD. The median plane of the cranium was used to mirror the healthy side of the cutaneous model onto the defect side. This Mirrored Skin 3D Information was then used to design the Outer Contour of the HPPD. An example of a unique HPPD is shown in Fig 1.

Finally, the 3D design was furnished with Ventilation Holes and subject-specific Slots indicated digitally by the study team and dimensionally compatible with the horizontal frontal and vertical chin strap, hand-sewn or press-studded elastic fabric bands. The bands enabled the subject to don and doff the HPPD with or without minimal assistance. The final 3D design was then sent to Fortus 450mc (Stratasys, USA) and fabricated in FDM Nylon 12 (Stratasys, USA) with 0.1778mm layer height, Infill Density 33% Hexagram, and 1-3mm Body Thickness (GrabCAD Print, Stratasys, USA). (10202002777Q, National Healthcare Group: 2019047) The single material which the helmets were made of (FDM Nylon 12, Stratasys, USA) was mentioned in line 224. This material was selected for its printability at a fine resolution

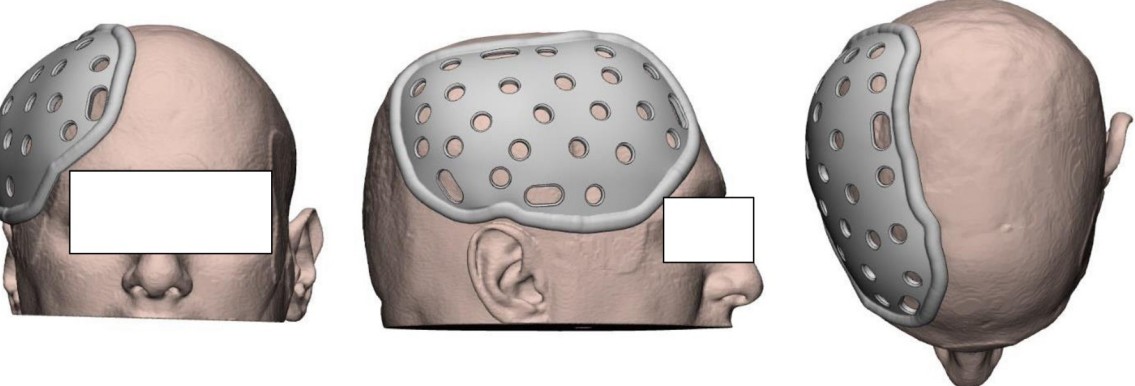

**Fig 1. 3-Dimensional rendering of the head protection prototype device (HPPD).** The rendering excludes the cranium model and the elastic fabric bands. Grey represents HPPD. Beige represents cutaneous model. (Image courtesy of AuMed, Singapore). Anterior (left image), Right lateral (centre image), Top (right image).

(0.1778mm) for the available machine (Fortus 450mc), its high Elongation at Break (30%, ASTM D638, Stratasys USA), low specific gravity (1.00, ASTM D648, Stratasys USA) and comparatively high tensile strength (4,600 psi, ASTM D638, Stratasys, USA). Screening was made across materials printable at the same resolution and machine. Duration to design, fabricate, post-processing and delivery of each HPPD took 7 days or less.

**(ii) Fitting of HPPD.** The HPPD was fitted either in the ward or clinic and subjects were monitored for the first 30 minutes for (i) symptoms of pain, pressure, discomfort, or (ii) wound changes of redness or wound dehiscence. Where there was incomplete apposition between the skin and HPPD, self-adhesive poly-cushion padding was used to provide additional support. (https://www.performancehealth.com/rolyan-polycushion-padding#sin= 93702) Upon feasible fitting for the first 30 minutes with HPPD, subjects' carers were instructed on; (i) donning and doffing techniques, (ii) observation of skin condition, (iii) cleansing the HPPD with a wet wipe, (iv) progressive wearing schedule advised during ambulation, outdoor activities; and (v) daily written logging of their wearing schedules. The duration of wear commenced at 15 minutes twice daily with wound monitoring, and increased progressively by 15–30 minutes daily, till ~2–3 hours of continuous wear per session were tolerated. Removal and skin rest was advised after 2–3 hours of wear to prevent skin breaks. Subjects and carers were then discharged with their HPPD with written care instructions, wearing schedules, skin monitoring reminders and a phone number of the study coordinator in case of queries.

**(iii) Follow-up phase over 8 weeks.** Phone follow ups (FU) were conducted at days 1,3 and, week 6 post-fitting and face to face FU visits were conducted at 1,2,4, and 8 weeks after HPPD fitting by non-blinded study assessors. At each of the visits, objective and subjective outcomes were recorded; (i) wearing time/day—WTD in absolute hours by subjects' log, (ii) types of activities performed during HPPD wear, (iii) presence /absence of pain, pressure, pruritus (rating scale absolute units 0–10); (iv) presence / absence of skin changes, dislodgement during wear and; (v) subjective ratings of cosmesis (rating scale absolute units 0–10), higher scores reflecting better ratings. Subjects were allowed to retain their HPPDs beyond 8 weeks FU if deemed safe by study team, as approved by ethical boards. (S1 and S3 Files).

**(iv) Study outcomes.** The primary outcome was the percentage of subjects who were safety fitted without pain reports or wound changes within 30 minutes of HPPD fitting.

The secondary outcomes included subjective and objective outcomes monitored by the study team over 8 weeks of follow-up (FU).

Baseline demographic, injury-related, DC details, acute and rehabilitation length of stay, clinical rehabilitation outcomes using the Functional Independence Measure (FIM), ranging from 18–126; and total, subset motor (total 91) and cognitive scores (total 35) were recorded [19].

## (V) Statistical analysis

This study was shaped to be a small-scale preliminary study mainly to evaluate the feasibility and safety of HPPD thus a large number of subjects was not recommended before the safety of the device is ascertained. Hence, the sample size of 10 was decided on and the outcomes were mainly descriptive. Data regarding non compliers or drop outs were not replaced by last recorded variables. Statistical analyses were performed with the Statistical Product and Service Solutions version 27 program. Descriptive statistics were used to characterise demographic, clinical, HPPD, follow-up and rehabilitation data. Categorical variables were displayed with frequencies and percentages. The distribution of numerical data was assessed using skewness, kurtosis and histogram. Differences in Total, Motor and Cognitive FIM between admission and discharge were tested with Wilcoxon's Signed-Ranked test. Statistical significance was denoted as $P < 0.05$.

## Results

### (i) Subject recruitment flow and primary outcome measure

The study protocol and ethics approvals allowed for up to 2 HPPD fabrications per subject in case of misplacement or technical errors. There were no reports of protocol deviations during the 8 week study follow-up period. The subject recruitment flow process is shown in Fig 2.

All 10 (100%) enrolled subjects were successfully fitted with HPPD during their initial visit and all tolerated 30 minutes of wear without pain, pressure or incisional / surrounding wound changes, achieving the study's primary outcome. All 10 (100%) subjects also progressed to the 8 weeks FU phase. One drop-out (subject 2, 10%) occurred at week 4 FU due to an unrelated medical hospitalisation. Two subjects (subjects 3, 10, 20%) received CP during the 8 week HPPD FU, thus discontinued HPPD and the remaining 7 (70%) subjects completed 8 weeks FU.

### (ii) Baseline data regarding study subjects, clinical, rehabilitation and HPPD characteristics

The baseline individual demographic, clinical, rehabilitation and HPPD characteristics of sample are shown in Table 1.

A total of 12 HPPDs were fabricated for 10 enrolled subjects as 2 (subjects 5, 6) each received 2 HPPDs, due to a misplacement and a suboptimal fit each. Of the 10 subjects, 50% were male (5/10), 40% (4/10) were of Chinese ethnicity and the mean (SD) age was 46 (14) years. Eight in 10 subjects (80%) had right-sided unilateral DC, 2 with left-sided; and DC aetiologies included, traumatic brain injury (TBI) (4), intracerebral haemorrhage (ICH) (3), acute ischaemic stroke (AIS) (2), and subarachnoid haemorrhage (SAH) (1). The mean (SD) duration post-DC to HPPD fitting was 110 (76) days, median 88 days, range 45–311 days. Six subjects (60%) were fitted with HPPD during their inpatient rehabilitation while 4 subjects (40%) were fitted in the outpatient rehabilitation phase. With the exception of 3 subjects (subjects 2, 3, 10), the remaining 7 (70%) needed physical assistance from carers to don and doff

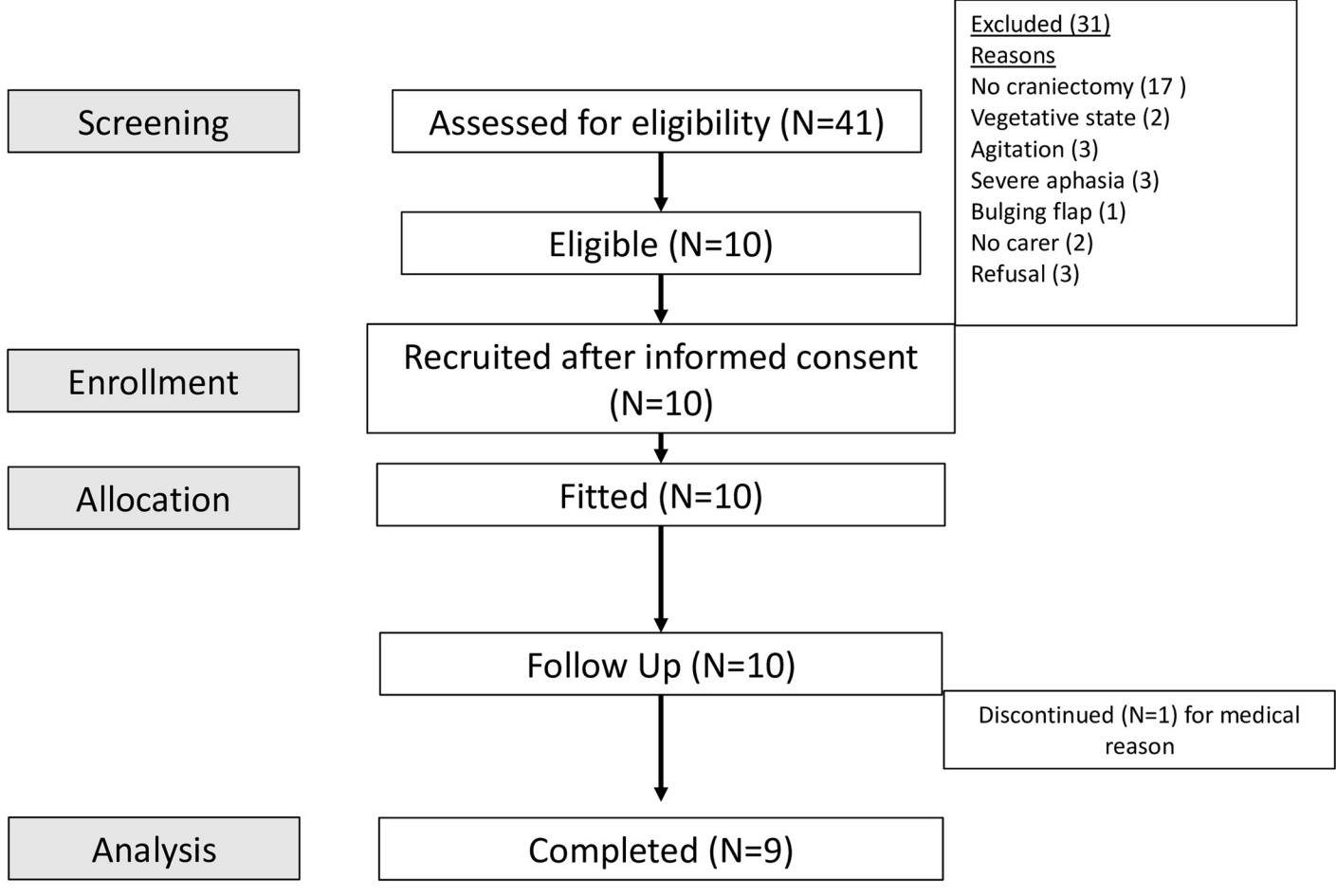

**Fig 2. Subject recruitment flow diagram.**

their HPPDs, due to hemiparetic arms. The final weights of HPPDs without straps ranged from 37 to 100g, mean (SD) of 61.2g (19.88).

Summary statistics of the patients and HPPD were presented in Table 2.

Summary statistics of FIM data on admission and discharge from rehabilitation are shown in Table 3, and this showed significant gains in admission compared to discharge total, subset motor and cognitive discharge FIM (Table 3).

The individual's pre and post HPPD fitting images are shown in Fig 3.

The individual subjects' and mean wearing times /day (WTD) in hours (absolute numbers) are shown in Fig 4.

The individual subjects' and mean cosmesis ratings in absolute numbers (0–10) are shown in Fig 5.

For the 7 subjects who completed 8 weeks FU, WTD ranged from 2 to 4 hours per day. Two subjects recorded ~ 8 hours of wear with intermittent rest breaks, one worked via tele-conferencing whereby cosmesis was vital for her self-image, and another was community ambulant. Subject 5, who recorded 0 WT/hours post-discharge, was severely hemiplegic and had insufficient physical assistance to don/doff HPPD. Overall cosmesis ratings (range 0–10/10) were favorable (mean 6/10) and tended to increase towards FU week 8.

**Table 1. Baseline individual demographic, clinical and HPPD characteristics (N = 10).**

| Subject No | Age (Years) | Gender/ Race | Cause of DC | Side DC | Acute LOS (Days) | Rehab LOS (Days) | Days To Fit | FIM Admit / Discharge | Weight HPPD (g) | Completed study /Week | HPPD retained after week 8 |
|---|---|---|---|---|---|---|---|---|---|---|---|
| 1 | 65 | F/Chinese | TBI | Right | 24 | 45 | 117 | 43/100 | 52 | Yes/8 | Yes |
| 2 | 60 | F/Malay | TBI | Right | 15 | 48 | 311 | NA/NA | 46 | No/2 | NA |
| 3 | 45 | F/Filipino | TBI | Right | 22 | 82 | 112 | 87/109 | 50 | Yes/4 | No** |
| 4 | 27 | M/Malay | TBI | Left | 68 | 133 | 140 | 76/91 | 100 | Yes/8 | Yes |
| 5* | 24 | M/Chinese | CVA-INF | Right | 22 | 47 | 66 | 48/90 | 79 | Yes/8 | Yes |
| 6* | 51 | F/Chinese | CVA-ICH | Right | 15 | 59 | 94 | 24/75 | 51 | Yes/8 | Yes |
| 7 | 39 | M/Malay | CVA-ICH | Right | 66 | 52 | 81 | 52/75 | 37 | Yes/8 | Yes |
| 8 | 63 | F/Malay | CVA-ICH | Right | 15 | 79 | 76 | 20/30 | 84 | Yes/8 | Yes |
| 9 | 47 | M/Chinese | CVA-INF | Left | 17 | 81 | 60 | 26/72 | 52 | Yes/8 | Yes |
| 10 | 37 | M/Indian | CVA-SAH | Right | 26 | 32 | 45 | 64/94 | 61 | Yes/4 | No** |

HPPD: Head Protection Prototype Device; DC: Decompressive Craniectomy; F: Female; M: Male; TBI: Traumatic Brain Injury; CVA: Cerebrovascular Accident; INF: Infarction; ICH: Intracerebral Haemorrhage; LOS: Length of Stay, FIM: Functional Independence Measure; NA: Not available

*Received 2 HPPD each.

** Received cranioplasty prior to week 8.

### (iii) Adverse events during 8 weeks follow-up period

Summary statistics of adverse events over 8-week FU are shown in Table 4.

Adverse events unrelated to HPPD-wear were recorded in 4 /10 (40%) subjects. None of these were deemed to be directly related to HPPD wear. No falls were reported.

No HPPD-related incisional wound changes or dehiscence were observed during the 8-week FU period. Two of 10 subjects (20%) had no subjective complaints of pain, pressure, itch, dislodgement or skin changes during 8 weeks FU. The remaining 8/10 (80%) had various transient complaints related to HPPD-related wear. (Table 4) Complaints of pressure were the most prevalent in 60% (6/10) and this was of mild-moderate degree, graded on Likert scale (0–10) at 3/10–5/10, followed by pruritus at the frontal bone margin in 40% (4/10, range 3/10–7/10), followed by pain in 20% (2/10, range 2/10–4/10). Two subjects had transient and self-limiting frontal skin imprints, observed at week 1 and 2 FU respectively, possibly contributed by prolonged wear for 4–5 hours and tight-fitting caps worn over HPPD.

## Discussion

To our knowledge, this is the largest clinical study to date, describing the feasibility, safety and usability of external cranial helmet prototype in a sample of DC patients during rehabilitation. Findings from this study showed preliminary feasibility, safety and acceptability of the HPPD for temporary cranial protection. The primary study outcome of safe and skin-tolerated HPPD fitting was achieved in 100% of subjects. For secondary outcomes, no serious HPPD-related adverse events relating to severe pain, skin dehiscence or infection were reported for all 10 subjects during 8-weeks FU. A single dropout after week 4 FU was due to unrelated medical reasons.

All HPPDs fabricated during the inpatient rehabilitation phase were worn during rehabilitation therapies, which resolved the problem of a lack of customized protective helmets.

During 8 weeks of FU, self-reported complaints such as pressure, mild-moderate pain and itch/pruritus were prevalent in 80% (8/10) and these were transient being relieved by rest from

**Table 2. Summary of baseline demographic, clinical and HPPD characteristics.**

| Variables | N | Outcome | | |
|---|---|---|---|---|
| | | Mean (SD) | Median (IQR) | Frequency (%) |
| **Demographic characteristics** | | | | |
| Age in years | 10 | 46 (14) | 46 (37, 60) | |
| Gender | 10 | | | |
| Male | | | | 5 (50.0) |
| Female | | | | 5 (50.0) |
| Race | 10 | | | |
| Chinese | | | | 4 (40.0) |
| Non-Chinese | | | | 6 (60.0) |
| **Clinical Characteristics** | | | | |
| Aetiology | 10 | | | |
| TBI | | | | 4 (40.0) |
| CVA-AIS | | | | 2 (20.0) |
| CVA-ICH | | | | 3 (30.0) |
| CVA-SAH | | | | 1 (10.0) |
| Side of DC | 10 | | | |
| Right | | | | 8 (80.0) |
| Left | | | | 2 (20.0) |
| Days to Cranioplasty* | 4 | 252 (158) | 271.5 (118.0, 386.0) | |
| Acute LOS in days | 10 | 29 (20) | 22 (15, 26) | |
| Rehab LOS in days* | 10 | 66 (29) | 56 (47, 81) | |
| Discharge Destination | 10 | | | |
| Home | | | | 8 (80.0) |
| Community Home | | | | 2 (20.0) |
| **HPPD Characteristics** | | | | |
| Days to Fitting HPPD* | 10 | 110 (76) | 88 (66, 117) | |
| Weight without straps in gram | 10 | 61.20 (19.88) | 52.0 (50.0, 79.0) | |
| Weight with straps in gram | 10 | 69.00 (19.36) | 61.0 (55.0, 79.0) | |
| HPPD retained post-week 8** | 9 | | | |
| No | | | | 2 (22.22) |
| Yes | | | | 7 (77.78) |
| **FIM scores**** | | | | |
| Admission FIM | | | | |
| Total | 9 | 48.89 (23.56) | 48.0 (26.0, 64.0) | |
| Motor | 9 | 32.00 (17.05) | 29.0 (22.0, 42.0) | |
| Cognitive | 9 | 15.56 (6.82) | 20.0 (11.0, 21.0) | |
| Discharge FIM | | | | |
| Total* | 9 | 81.78 (23.03) | 90.0 (75.0, 94.0) | |
| Motor | 9 | 54.00 (19.56) | 54.0 (46.0, 65.0) | |
| Cognitive | 9 | 25.22 (5.89) | 27.0 (21.0, 29.0) | |

HPPD: Head Protection Prototype Device, DC: Decompressive Craniectomy, TBI: Traumatic Brain Injury, CVA: Cerebrovascular Accident, INF: Infarction, ICH: Intracerebral Haemorrhage; LOS: Length of Stay, FIM: Functional Independence Measure.

*: Data were skewed.

**: Data not available due to 1 drop out.

**Table 3. Changes in total, motor and cognitive FIM between admission and discharge.**

|  | Admission | Discharge | FIM gain | P value |
|---|---|---|---|---|
| **Total FIM** | 48.0 (26.0, 64.0) | 90.0 (75.0, 94.0) | 30.0 (18.5, 48.5) | 0.008 |
| **Motor FIM** | 29.0 (22.0, 42.0) | 54.0 (46.0, 65.0) | 21.0 (10.0, 31.5) | 0.008 |
| **Cognitive FIM** | 20.0 (11.0, 21.0) | 27.0 (21.0, 29.0) | 9.0 (5.5, 14.0) | 0.012 |

Wilcoxon's Signed-Ranked test.

the HPPD; while WTD was consistent and subject-reported cosmesis ratings were generally positive over 8 weeks.

### (i) Characteristics of HPPDs

The weight of the fabricated HPPDs ranged from 37-100grams, (Table 2) in comparison to standard rigid ventilated helmets (250-350grams). The largest skull defect in a subject with a 100g HPPD fitted well with no complaints over 8 weeks.

### (ii) Baseline characteristics of subjects

The wide age range of subjects (24–65 years) were attributable to the inclusion of both stroke and TBI subjects and 75% of 4 TBI subjects were older than usual TBI populations, being > 40

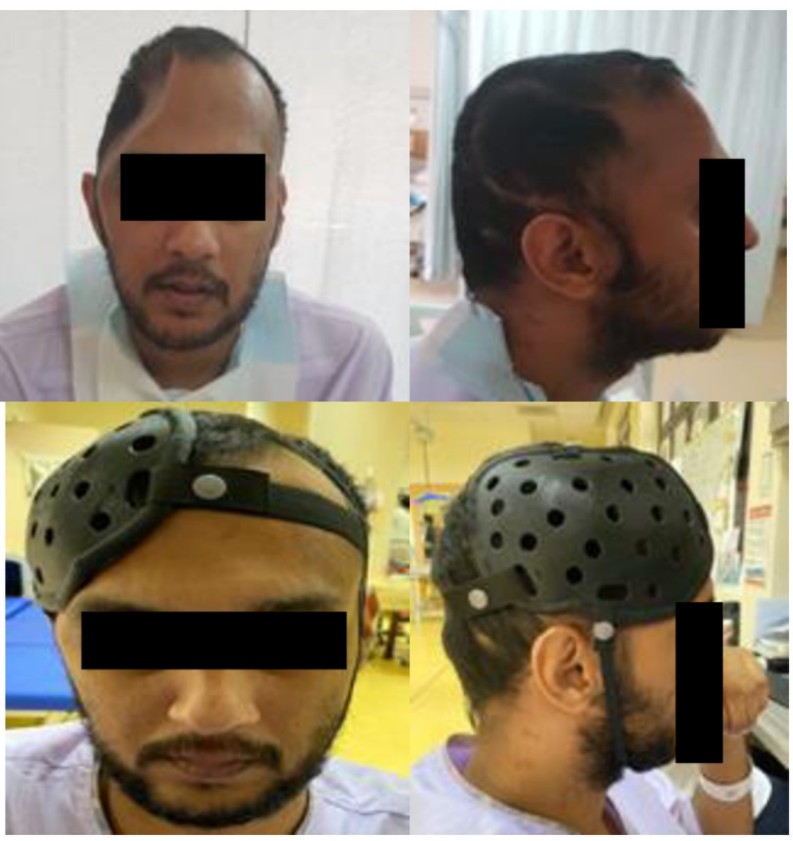

**Fig 3. Subject pre and post head protection prototype device (HPPD) fitting showing anterior (left) and right lateral (right) views.**

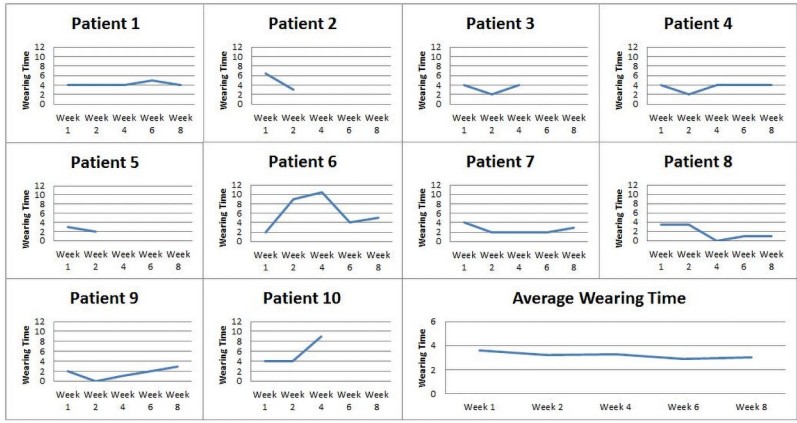

**Fig 4. Individual and group wearing time (hours/day) by time point (weeks).** Subjects 1,4,5,6,7,8,9 completed 8 weeks FU. Subject 2 dropped out at week 4 FU. Subjects 3, 10 received cranioplasties prior to week 8 FU. Subject 5, who was institutionalized after week 2 FU had no assistance to don HPPD. The average wearing time was computed based on complete data available (N = 6).

years while all 5 enrolled stroke subjects were relatively younger than usual stroke populations, at < 63 years. (Tables 1 and 2) There was a disproportionate number of females (50%, 5/10) in stroke and TBI populations, as well as a predominance of non-Chinese subjects (60%, 6/10) overall in the sample; in comparison to the local population's 76% Chinese racial predominance [20–24]. These findings reflected significant study bias related to the small sample size, as well as clinician selection prior to inpatient admission, where by older and frail patients were admitted to community hospitals rather than rehabilitation hospitals.

### (iii) HPPD characteristics, wearing time and cosmesis ratings

The low density /mass of the HPPD and ventilatory spaces were likely contributing factors to safe initial fitting without pain and skin breakdown and during subsequent 8 weeks FU period. Mild pain was noted in 20% and 20% had transient frontal, non-incisional skin imprints

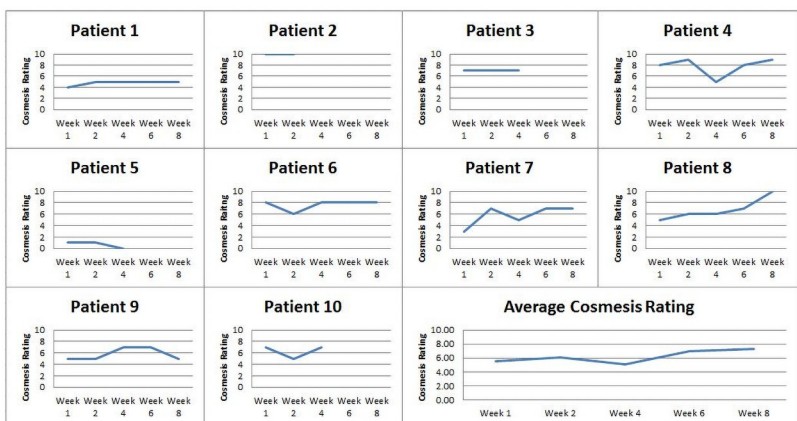

**Fig 5. Individual and group cosmesis rating (0–10) by time point (weeks).** Subjects 1,4,5,6,7,8,9 completed 8 weeks FU. Subject 2 dropped out at week 4 FU. Subjects 3, 10 received cranioplasties prior to week 8 FU. Subject 5, institutionalized after week 2 FU had no assistance to don HPPD. The average wearing time was computed based on complete data available (N = 6).

**Table 4. Summary of types and number of reported adverse events (N = 10).**

| Subject code | Related to HPPD | | | | | Non-Related to HPPD | | |
|---|---|---|---|---|---|---|---|---|
| | Pain | Pressure | Itch or rash | Dislodgement | Skin changes | Focal Seizures | Syncope | Others |
| 001 | No | No | Yes/5 | No | Yes/1[1] | No | No | No |
| 002 | No | Yes/2 | No | No | No | No | No | Yes /1[2] |
| 003 | No | Yes/1 | Yes/3 | No | No | No | No | No |
| 004 | No | No | No | No | No | No | No | No |
| 005 | No | No | Yes/1 | No | No | No | No | No |
| 006 | No | No | No | No | No | Yes/1 | No | No |
| 007 | Yes/1 | Yes1 | Yes/1 | No | No | No | No | No |
| 008 | No | Yes/2 | No | No | No | No | Yes/1 | No |
| 009 | No | Yes/7 | No | No | No | Yes/1 | No | No |
| 010 | Yes/1 | Yes/1 | No | No | Yes/1[1] | No | No | No |

Cases are reported as: (Yes/No) / (Number of reported adverse events).

[1] Transient imprints.

[2] Hospitalization related to medical illness.

related to prolonged wearing time, concomitant use of headwear and low subcutaneous fat padding in the frontal bone area.

The consistent mean total WTD of 3–4 hours /day (Fig 3) reflected a variety of factors; the study protocol of recommended HPPD wear during locomotor activities, timed removal after 2–3 hours of wearing time to allow skin rest, independence levels to for donning and doffing and availability of carers. Unique individual WTD variations noted in 2 female subjects (subjects 2 and 6) reported 8 hours /day WTD, attested to acceptable levels of comfort and tolerability. In contrast, low WTD in a single subject was due to social factors, being in a nursing home. In terms of cosmesis, the ratings were consistently positive throughout the study duration indicating levels of acceptability in both genders (Fig 4).

Our findings concurred with case reports of customized 3D printed helmets improving cosmesis and socialization in a patient with bifrontal craniectomy compared to standard off-the shelf helmets, and augmenting safety for rehabilitation exercises in a case of pediatric congenital acrania, who was not yet a surgical candidate [25, 26].

**Adverse events related to HPPD.** Notably, no wound dehiscence was noted, while self-limited mild-moderate pressure, pruritus and pain were prevalent in 80% (8) of subjects predominantly at the frontal bone areas. (Table 4) Several reasons were postulated; frontal bony regions, were relatively unprotected by hair which served as additional buffering and post-operative scar sensitivity to external pressure from HPPD.

## (iv) Study limitations

We highlight the following study limitations: (i) the small number of subjects, (ii) single centre descriptive study, (iii) lack of a control group, (iv) short follow-up duration of 8 weeks, (v) exclusion of bifrontal craniectomy subjects, (vi) lack of data related to quality of life, and (vii) lack of carers' perspectives.

## (v) Future work

This feasibility study also highlighted unique challenges in terms of; the need for CT stereo-taxis images as a HPPD pre-design requirement, prevalence of complaints related to transient

skin interface comfort, and a week-long duration for design and fabrication. Further development would include; (i) possibility of 3D-rendering via 3-dimensional camera imaging instead of CT images, (ii) direct impact tests on strength and safety of HPPD against cranial flap injury, (iii) composite materials and/or larger margin off-sets to improve skin interface comfort, (iv) semi-rigid integrated frontal attachments and flexible chin straps to facilitate one-handed donning in hemiplegic patients, (v) shorter printing time, and (vi) preliminary cost analysis studies.

## Conclusions

In conclusion, findings from this pilot cohort, demonstrate the preliminary feasibility, safety and usability for a 3D-printed HPPD as an acceptable external cranial protection post-DC over 8 weeks of follow-up. Supervision, monitoring and regular rest breaks during HPPD wear remained important to prevent secondary cranial flap skin complications in this vulnerable population. This study may inform healthcare professionals of new 3DP application potentials for post-DC and awaits reproducibility in a larger cohort with a control group using usual helmets. Overall, cheaper 3D-printing costs compared with standardized methods would be potentially advantageous for widespread application.

## Supporting information

**S1 File. Data collection form (line 258).**
(PDF)

**S2 File. Patient informed consent for publication of image.**
(PDF)

**S3 File. Study protocol.**
(PDF)

**S4 File. MS Excel file of de identified data points.**
(XLS)

**S1 Checklist.**
(PDF)

## Acknowledgments

We thank Ms Hong Li, Jiao for her clinical trial assistance and all subjects and their carers for their participation.

## Author Contributions

**Conceptualization:** Karen Sui Geok Chua, Rathi Ratha Krishnan, Tegan Kate Plunkett, Catherine M. Chia, Jun Cong Looi, Jai Rao.

**Data curation:** Karen Sui Geok Chua, Rathi Ratha Krishnan, Jia Min Yen, Tegan Kate Plunkett, Yan Ming Soh, Chien Joo Lim, Catherine M. Chia, Jun Cong Looi, Suan Gek Ng, Jai Rao.

**Formal analysis:** Karen Sui Geok Chua, Rathi Ratha Krishnan, Jia Min Yen, Chien Joo Lim, Jai Rao.

**Funding acquisition:** Karen Sui Geok Chua, Rathi Ratha Krishnan, Tegan Kate Plunkett.

**Investigation:** Karen Sui Geok Chua, Rathi Ratha Krishnan, Jia Min Yen, Tegan Kate Plunkett, Yan Ming Soh, Chien Joo Lim, Catherine M. Chia, Jun Cong Looi, Jai Rao.

**Methodology:** Karen Sui Geok Chua, Rathi Ratha Krishnan, Jia Min Yen, Tegan Kate Plunkett, Yan Ming Soh, Chien Joo Lim, Catherine M. Chia, Jun Cong Looi, Suan Gek Ng, Jai Rao.

**Project administration:** Karen Sui Geok Chua, Rathi Ratha Krishnan, Jia Min Yen, Tegan Kate Plunkett, Yan Ming Soh, Catherine M. Chia, Suan Gek Ng.

**Resources:** Karen Sui Geok Chua, Tegan Kate Plunkett, Yan Ming Soh, Catherine M. Chia, Jun Cong Looi, Suan Gek Ng.

**Software:** Chien Joo Lim, Catherine M. Chia, Jun Cong Looi.

**Supervision:** Karen Sui Geok Chua, Rathi Ratha Krishnan, Jia Min Yen, Tegan Kate Plunkett, Yan Ming Soh, Catherine M. Chia, Jun Cong Looi, Suan Gek Ng, Jai Rao.

**Validation:** Karen Sui Geok Chua, Tegan Kate Plunkett, Chien Joo Lim, Catherine M. Chia, Jun Cong Looi.

**Visualization:** Catherine M. Chia, Jun Cong Looi.

**Writing – original draft:** Karen Sui Geok Chua, Chien Joo Lim, Catherine M. Chia.

**Writing – review & editing:** Karen Sui Geok Chua, Rathi Ratha Krishnan, Jia Min Yen, Tegan Kate Plunkett, Yan Ming Soh, Chien Joo Lim, Catherine M. Chia, Jun Cong Looi, Suan Gek Ng, Jai Rao.

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
