## [Decision Letter · Decision Letter 0]

30 Jul 2021

PONE-D-20-38286

3D-printed external cranial protection following decompressive craniectomy after brain injury: An exploratory clinical trial

PLOS ONE

Dear Dr. Chua,

Thank you for submitting your manuscript to PLOS ONE. After careful consideration, we feel that it has merit but does not fully meet PLOS ONE’s publication criteria as it currently stands. Therefore, we invite you to submit a revised version of the manuscript that addresses the points raised during the review process.

We look forward to receiving your revised manuscript.

Kind regards,

Luigi Maria Cavallo

Academic Editor

PLOS ONE

Journal Requirements:

2. Thank you for submitting your clinical trial to PLOS ONE and for providing the name of the registry and the registration number. The information in the registry entry suggests that your trial was registered after patient recruitment began. PLOS ONE strongly encourages authors to register all trials before recruiting the first participant in a study.

1) your reasons for your delay in registering this study (after enrolment of participants started);

2) confirmation that all related trials are registered by stating: “The authors confirm that all ongoing and related trials for this drug/intervention are registered”.

3. During our internal checks, we noted that one or more of the authors is affiliated with AuMed Pte Ltd. Please add this affiliation information to the Financial Disclosure/Conflict of Interest statement in your online submission form.

4. Please note that PLOS ONE does not copy edit accepted manuscripts (https://journals.plos.org/plosone/s/criteria-for-publication#loc-5). To that effect, please ensure that your submission is free of typos and grammatical errors.

*Please also consider whether some images in your submission should be moved to the Supplemental information due to a potentially graphic nature.

6. Please upload a new copy of Figure 4 and 5 as the detail is not clear. Please follow the link for more information: https://blogs.plos.org/plos/2019/06/looking-good-tips-for-creating-your-plos-figures-graphics/" https://blogs.plos.org/plos/2019/06/looking-good-tips-for-creating-your-plos-figures-graphics/.

Reviewers' comments:

Reviewer's Responses to Questions

**Comments to the Author**

1. Is the manuscript technically sound, and do the data support the conclusions?

Reviewer #1: Partly

Reviewer #2: Partly

Reviewer #3: Yes

2. Has the statistical analysis been performed appropriately and rigorously? 

Reviewer #1: Yes

Reviewer #2: Yes

Reviewer #3: I Don't Know

3. Have the authors made all data underlying the findings in their manuscript fully available?

Reviewer #1: Yes

Reviewer #2: No

Reviewer #3: Yes

4. Is the manuscript presented in an intelligible fashion and written in standard English?

Reviewer #1: Yes

Reviewer #2: Yes

Reviewer #3: Yes

5. Review Comments to the Author

Reviewer #1: In this work, the authors prospectively enrolled 10 craniectomy patients in a pilot trial of 3d printed HPPD as an alternative to standard helmet wearing. The work is novel, and analysis of findings based on their data is largely sound. I do have several concerns regarding the work that require clarification

1. The fundamental argument seems to be proposing an alternative to helmet wearing. However, other than in weight, there is no comparison to helmets (i.e. associated incisional breakdown, patient compliance, etc). Further, in the introduction, the authors make several claims regarding locally available helmet option shortcoming, but these require citation or the caveat of being authors anecdotal experience. I think it would also be important to compare as much as is able wearing patterns between HDDPs and helmets.

2. The authors note that HPPDs were not administered until a minimum of 30 days after hospital discharge. Was there helmet or other protective strategies made until then? And the authors note a mean of 76 days between discharge and HDDP application; How does this compare to typical availability of helmets? Does that mean that HDDPs were being donned on significantly more healed wounds than helmets?

3. In inclusion criteria, vegetative state requires clarification

4. In discussion, the discussion of FIM mean gains in a random selection of 10 craniectomy patients I do not think adds to the understanding of HDDP usage and should be removed

5. Was care taken to avoid contact with the incisions? Was pruritis or pressure in any patients observed along the incision? How much overlap with non-resected bone was made to allow for stable HDDP support?

Reviewer #2: Interesting study of custom printed external protection device in 10 patients s/p decompressive craniectomy. The authors state that the device is feasible and safe. I applaud the authors for their novelty in study design. I have several general and methodological comments for consideration.

- The authors state this was an "exploratory clinical trial" however there are no comparison groups or randomization. Possibly a cohort study is more appropriate?

- How were the subjects selected - convenience sample, consecutive enrollment over some period of time, ?

- Methodological issues: How far did the implant overlay existing bone and how was this optimized? What algorithm was used for 3-D printing? What material was the helmet made of and has this type of material been studied and or vetted? Was there any standardization of when and how long the helmets were worn? Why were there 12 helmets for 10 subjects? How did the authors determine 10 subjects to be a power sample sufficient for publishing on a clinical trial? Are there inherent weaknesses in the material comprising the helmet (e.g. to elements, trauma)? Were these patients considered for cranioplasty during the first 8 weeks? What comprised the cushion on the edges of the helmet to bone to prevent pressure injuries? Are long-term outcomes planned and/or evaluated?

- Without a comparison group, the results are observational rather than conclusive. How do the results compare to standard recovery for patients who did not have a helmet?

- In the conclusions the authors state that the HPPD was an "acceptable external cranial protection post-DC over 8 weeks". How do the authors qualify "acceptable external cranial protection"?

I think the paper has potential to be publishable, however in its current state is limited by very small sample size and unclear methodology and at best would qualify as a brief report or letter to the editor.

Reviewer #3: Important note: This review pertains only to ‘statistical aspects’ of the study and so ‘clinical aspects’ [like medical importance, relevance of the study, ‘clinical significance and implication(s)’ of the whole study, etc.] are to be evaluated [should be assessed] separately/independently. Further please note that any ‘statistical review’ is generally done under the assumption that (such) study specific methodological [as well as execution] issues are perfectly taken care of by the investigator(s). This review is not an exception to that and so does not cover clinical aspects {however, seldom comments are made only if those issues are intimately / scientifically related & intermingle with ‘statistical aspects’ of the study}. Agreed that ‘statistical methods’ are used as just tools here, however, they are vital part of methodology [and so should be given due importance].

COMMENTS: In my opinion, since it is clarified by authors that “The authors further investigate the feasibility, safety and usability of such a parameterized approach in its application towards external customized protection, head potection prototype device (HPPD) for post-acute decompression craniectomy (DC) patients.”, the word ‘feasibility‘ {Or a ‘pilot’ (as said in section ‘Materials and Methods - (I) Study Design)} may please be included along with the present ‘Exploratory’ [An exploratory clinical trial]. As is well known, a pilot study is a small-scale preliminary study conducted in order to evaluate feasibility, duration, cost, adverse events, and improve upon the study design prior to performance of a full-scale research project.

Note that ‘Exploratory’ studies termed pilot and/or feasibility studies, are a key step in assessing the feasibility and value of progressing to an effectiveness study. Such studies can provide vital information to support more robust evaluations, thereby reducing costs and minimising potential harms of the intervention [Hallingberg, B., Turley, R., Segrott, J. et al. Exploratory studies to decide whether and how to proceed with full-scale evaluations of public health interventions: a systematic review of guidance. Pilot Feasibility Stud 4, 104 (2018). https://doi.org/10.1186/s40814-018-0290-8].

I am sure that the authors are aware of the well-known drawbacks of a single-arm design [a type of Quasi-experimental research], and it is often said that ‘alright to have ‘single-arm design’ (before-after study) for pilot study or when that is the only possibility’, however, it is very essential to keep the limitations in mind while interpreting results. Further, note that a classical/ideal clinical trial/study needs/requires a concurrently {but similarly} handled/treated appropriately selected/chosen control/comparison parallel group/arm.

Design and other details [like Fabrication, etc.] of HPPD are alright/perfect. You may know that comparison of means from non-Gaussian distributions or when level of measurement may not ‘ratio’, for paired samples (before-after) [i.e. Two samples comparison – Paired Case] is preferred (over paired t-test) to be done by Wilcoxon’s Signed-Ranked test. Further, kindly note that presently ‘SPSS’ stands for ‘Statistical Product and Service Solutions’ [and not for Statistical Package for the Social Science any more]. Refer to page 608 of a text book ‘Medical Biostatistics’ by Indrayan & Sarmukaddam, ISBN: O-8247-0426-6, Marcel Dekkar Inc., New York, 2001.

Study Limitations [section: (iv)] are totally agreed and by statistical point of limitations 1 & 3 [(i) the small number of subjects, (ii) single centre descriptive study, (iii) lack of a control group] are causing to stop the review here. As pointed out in the ‘Important note’ above (in the beginning) this review pertains only to ‘statistical aspects’ of the study and so ‘clinical aspects’ [like medical importance, relevance of the study, ‘clinical significance and implication(s)’ of the whole study, etc.] are to be evaluated [should be assessed] separately/independently.

6. PLOS authors have the option to publish the peer review history of their article (what does this mean?). If published, this will include your full peer review and any attached files.

Reviewer #1: **Yes: **Michael J. Feldman

Reviewer #2: No

Reviewer #3: No

---

## [Author Response · Author response to Decision Letter 0]

3 Aug 2021

4 August 2021

Dear PLOS one Editors. 

RE: PONE-D-2038286 rebuttal 

On behalf of the authors, we thank you for the comprehensive comments to improve the quality and clarity of the manuscript and we submit a revised version with our point by point (>>) rebuttal. 

Journal Requirements:

https://imsva91-ctp.trendmicro.com:443/wis/clicktime/v1/query?url=https%3a%2f%2fjournals.plos.org%2fplosone%2fs%2ffile%3fid%3dwjVg%2fPLOSOne%5fformatting%5fsample%5fmain%5fbody.pdf&umid=3B088F6D-C85E-1D05-90DC-B99F42FB63A6&auth=6e3fe59570831a389716849e93b5d483c90c3fe4-2ae0558fb485b957534e90fb0a016cc60d768301 and 

https://imsva91-ctp.trendmicro.com:443/wis/clicktime/v1/query?url=https%3a%2f%2fjournals.plos.org%2fplosone%2fs%2ffile%3fid%3dba62%2fPLOSOne%5fformatting%5fsample%5ftitle%5fauthors%5faffiliations.pdf&umid=3B088F6D-C85E-1D05-90DC-B99F42FB63A6&auth=6e3fe59570831a389716849e93b5d483c90c3fe4-a6dca308f3906a6aa734d1bdce1537fc4809c02b

>> We have complied with the title page formatting suggestions accordingly. Acknowledgements have been moved from end of manuscript to the title page.

2. Thank you for submitting your clinical trial to PLOS ONE and for providing the name of the registry and the registration number. The information in the registry entry suggests that your trial was registered after patient recruitment began. PLOS ONE strongly encourages authors to register all trials before recruiting the first participant in a study.

1) your reasons for your delay in registering this study (after enrolment of participants started);

>>NCT application was drafted after DSRB approval prior to first subject recruitment but there could have been inadvertent delays in the final approval and release of NCT document to public domain (dated 7 Nov 19) resulting in this anomaly.

2) confirmation that all related trials are registered by stating: “The authors confirm that all ongoing and related trials for this drug/intervention are registered”.

>>>>We apologise for this oversight and have included his in line 151-2.

3. During our internal checks, we noted that one or more of the authors is affiliated with AuMed Pte Ltd. Please add this affiliation information to the Financial Disclosure/Conflict of Interest statement in your online submission form.

>>We have added the COI statement for Catherine Chia and Jun Cong, Looi both from AuMed Pte Ltd in the online submission portal.

4. Please note that PLOS ONE does not copy edit accepted manuscripts (https://imsva91-ctp.trendmicro.com:443/wis/clicktime/v1/query?url=https%3a%2f%2fjournals.plos.org%2fplosone%2fs%2fcriteria%2dfor%2dpublication%23loc%2d5&umid=3B088F6D-C85E-1D05-90DC-B99F42FB63A6&auth=6e3fe59570831a389716849e93b5d483c90c3fe4-45e78fe424e5003c48290a1eedb0d0db86a1f495). To that effect, please ensure that your submission is free of typos and grammatical errors.

>> We have put the manuscript through spell check

*Please also consider whether some images in your submission should be moved to the Supplemental information due to a potentially graphic nature.

>>We have obtained written informed consent from the subject featured which has been uploaded and any identifiable facial features have been occluded. The images are important to demonstrate the project outcome and feasibility. 

>> We are working with our institution on the accession numbers/DOI linking with the deidentified data file and will provide this in due course.

6. Please upload a new copy of Figure 4 and 5 as the detail is not clear. Please follow the link for more information: https://imsva91-ctp.trendmicro.com:443/wis/clicktime/v1/query?url=https%3a%2f%2fblogs.plos.org%2fplos%2f2019%2f06%2flooking%2dgood%2dtips%2dfor%2dcreating%2dyour%2dplos%2dfigures%2dgraphics%2f%22&umid=3B088F6D-C85E-1D05-90DC-B99F42FB63A6&auth=6e3fe59570831a389716849e93b5d483c90c3fe4-98fa42cf7784c7e66137e95477262f2ebb1d0f31
https://imsva91-ctp.trendmicro.com:443/wis/clicktime/v1/query?url=https%3a%2f%2fblogs.plos.org%2fplos%2f2019%2f06%2flooking%2dgood%2dtips%2dfor%2dcreating%2dyour%2dplos%2dfigures%2dgraphics%2f&umid=3B088F6D-C85E-1D05-90DC-B99F42FB63A6&auth=6e3fe59570831a389716849e93b5d483c90c3fe4-bc1149b48426822212c49002f8132262f372855d.

>> We have re attached revised figure 4-5,without changes to figure 1-3 or tables .

Please refer to the response to reviewers for the rebuttal

Thank you '

---

## [Decision Letter · Decision Letter 1]

24 Sep 2021

3D-printed external cranial protection following decompressive craniectomy after brain injury: A pilot feasibility cohort study

PONE-D-20-38286R1

Dear Dr. Chua,

We’re pleased to inform you that your manuscript has been judged scientifically suitable for publication and will be formally accepted for publication once it meets all outstanding technical requirements.

Kind regards,

Luigi Maria Cavallo

Academic Editor

PLOS ONE

Additional Editor Comments (optional):

Reviewers' comments:

Reviewer's Responses to Questions

**Comments to the Author**

1. If the authors have adequately addressed your comments raised in a previous round of review and you feel that this manuscript is now acceptable for publication, you may indicate that here to bypass the “Comments to the Author” section, enter your conflict of interest statement in the “Confidential to Editor” section, and submit your "Accept" recommendation.

Reviewer #2: All comments have been addressed

Reviewer #3: (No Response)

2. Is the manuscript technically sound, and do the data support the conclusions?

Reviewer #2: Yes

Reviewer #3: (No Response)

3. Has the statistical analysis been performed appropriately and rigorously? 

Reviewer #2: N/A

Reviewer #3: (No Response)

4. Have the authors made all data underlying the findings in their manuscript fully available?

Reviewer #2: Yes

Reviewer #3: (No Response)

5. Is the manuscript presented in an intelligible fashion and written in standard English?

Reviewer #2: Yes

Reviewer #3: (No Response)

6. Review Comments to the Author

Reviewer #2: The authors have satisfactorily addressed my comments from the review.

Reviewer #3: COMMENTS: All of the comments made on earlier draft(s) by me (and hopefully by other respected reviewers also) were/are attended. However, I request the authors to note that [regarding Wilcoxon’s Signed Ranked test instead of paired ‘t’ test] even if the results and/or P values are similar (line 284), appropriate test should be used always.

7. PLOS authors have the option to publish the peer review history of their article (what does this mean?). If published, this will include your full peer review and any attached files.

Reviewer #2: No

Reviewer #3: **Yes: **Dr. Sanjeev Sarmukaddam

---

## [Editor Report · Acceptance letter]

21 Oct 2021

PONE-D-20-38286R1 

3D-printed external cranial protection following decompressive craniectomy after brain injury: A pilot feasibility cohort study 

Dear Dr. Chua:

I'm pleased to inform you that your manuscript has been deemed suitable for publication in PLOS ONE. Congratulations! Your manuscript is now with our production department. 

Kind regards, 

on behalf of

Dr. Luigi Maria Cavallo 

Academic Editor

PLOS ONE